# Modified Clerodanes from the Essential Oil of *Dodonea viscosa* Leaves

**DOI:** 10.3390/molecules25040850

**Published:** 2020-02-14

**Authors:** Arnaud Marvilliers, Bertrand Illien, Emmanuelle Gros, Jonathan Sorres, Yoel Kashman, Hermann Thomas, Jacqueline Smadja, Anne Gauvin-Bialecki

**Affiliations:** 1Laboratoire de Chimie des Substances Naturelles et des Sciences des Aliments, Faculté des Sciences et Technologies, Université de la Réunion, 15 Avenue René Cassin, CS 92003, 97744 St Denis, Messag CEDEX 9, La Réunion, France; Arnaud.Marvilliers@univ-reunion.fr (A.M.); Bertrand.Illien@univ-reunion.fr (B.I.); emagros@orange.fr (E.G.); jacqueline.smadja@univ-reunion.fr (J.S.); 2Institut de Chimie des Substances Naturelles, CNRS, 91110 Gif-sur-Yvette, France; jon.sorres@gmail.com; 3School of Chemistry, Tel Aviv University, Tel Aviv 69978, Israel; kashman@tauex.tau.ac.il; 4Parc national de La Réunion, Secteur Nord, 165 allée des spinelles Bellepierre, 97400 St-Denis, France; hermann.thomas@reunion-parcnational.fr

**Keywords:** *Dodonea viscosa*, essential oil, furanoid (nor)diterpenes, bicyclo[5.4.0]undecane skeleton, electronic circular dichroism, density functional theory

## Abstract

*Dodonea viscosa* (L.) Jacq from Reunion Island (Indian Ocean) was investigated for its leaf essential oil composition. The plant was extracted by hydrodistillation and its essential oil analysed by gas chromatography coupled to mass spectrometry. This study revealed that oxygenated nor-diterpenes and diterpenes were one of the major chemical classes (> 50%) mainly consisting of three modified cyclopropylclerodanes containing a bicyclo[5.4.0]undecane ring system: one new furanoid norditerpene, dodovisate C, and two furanoid diterpenes, the known methyl dodovisate A and the new methyl iso-dodovisate A. These three compounds were isolated by liquid chromatography and their structures established on the basis of spectroscopic studies. The absolute configuration of dodovisate C was elucidated through a joint experimental and theoretical (B3LYP/6-311+G(d,p)) electronic circular dichroism study. The relative configurations of methyl dodovisate A and methyl iso-dodovisate A were determined using linear regressions of theoretical chemical shifts versus experimental values with the (B3LYP/6-311+G(d,p)) method.

## 1. Introduction

*Dodonea viscosa* is an ever-green shrub belonging to the family Sapindaceae with a cosmopolitan distribution in tropical, subtropical and warm temperate regions of Africa, India, Southern Asia, Australasia and South America. It is also met with throughout Reunion Island (Indian Ocean), often growing wild, but generally cultivated in the form of a hedge for ornamenting gardens, roads, etc. Moreover, preliminary observations of the wild populations, in various localities on Reunion Island, have shown a significant morphological diversity of the plant. Three main morphotypes can be distinguished according to the leaf shape: morphotype 1 (80–110 mm long, 15–25 mm wide), morphotype 2 (85–130 mm long, 10–13 mm wide) and morphotype 3 (120–150 mm long, 3–6 mm wide)

The plant possesses a great repute in folk medicine not only in India [1] Pakistan [2], Ethiopia [3,4], Tanzania [5], South Africa [6,7,8,9,10], Peru [11], Ecuador [12], Brazil [13], Mexico [14,15], but also in Madagascar [16], Mauritius [17,18], Rodrigues [19], and Reunion Island [20,21,22,23,24,25]. In Reunion Island, this plant is locally really appreciated for its healing virtues: it is considered to be very efficacious against arthrosis, gout, rheumatism, haemorrhoid, haematoma, asthma, sinusitis, and nephritic colic. It is also reputed to have hypertensive properties. Owing to its medicinal properties, this species has been registered in the French Pharmacopoeia since 2012 [26,27].

Since it is widely used in folk medicine, the plant has been extensively investigated for its biological and pharmacological properties [3,5,7,15,18,28,29,30,31,32,33,34,35,36,37,38,39,40,41,42,43,44], as well as its chemical composition. According to previous phytochemical studies, the species contains diterpenes, sapogenins, saponins, sterols, triterpenes, flavonoids and other phenolic compounds [37,44,45,46,47,48,49,50,51,52,53,54,55,56,57,58,59,60,61,62,63]. Concerning more specifically volatile compounds from *Dodonea viscosa*, only one study devoted to essential oil sample from Saudi Arabia has been carried out [64]. This study indicated that the leaf essential oil of the plant mainly contains monoterpenes and sesquiterpenes. According to this literature survey, it also appears that up to now the chemical composition of *D. viscosa* from Reunion Island, has not been investigated.

Stimulated by the numerous pharmacological actions of *Dodonea viscosa* and in order to contribute to a better knowledge of this species growing on Reunion Island under several morphotypes, a detailed GC-MS examination of the leaf essential oil of *Dodonea viscosa* belonging to the group of morphotype 2 was undertaken. The present study deals with the isolation and structure elucidation of three modified cyclopropylclerodanes containing a bicyclo[5.4.0]undecane nucleus and present in high amount in the essential oil: one new furanoid norditerpene, dodovisate C (**1**), and two furanoid diterpenes, the known methyl dodovisate A (**2**) and the new methyl iso-dodovisate A (**3**). (Figure 1).

## 2. Results and Discussion

This section is divided into two parts. The first one, is devoted to the detailed analysis of the essential oil by GC-MS. The second one describes the isolation and identification by means of 1D and 2D NMR spectroscopy of the three oxygenated compounds, one nor-diterpene (dodovisate C (**1**)) and two diterpenes (methyl dodovisate A (**2**) and methyl iso-dodovisate A (**3**)) present in high amounts in the essential oil. In this part, computational studies applied to dodovisate C in order to determine its absolute configuration, and to methyl dodovisate A and methyl iso-dodovisate A in order to determine their relative configuration, are described.

### 2.1. Essential Oil Composition

The essential oil obtained by hydrodistillation of the leaves of D. viscosa, exhibited a pale yellow to yellow-green colour, a strong odour and a partially solid consistency. The oil yield, calculated from fresh material, was 0.10%. The chromatographic analyses (GC-MS) of the essential oil sample allowed the detection of more than 80 compounds accounting for 80.5% of the total oil composition. Their retention indices and their composition are listed in Table 1. All the constituents grouped by chemical classes are arranged according to their elution order on the SPB-5 column.

The essential oil composition was characterized by a high amount of oxygenated nor-diterpenes and diterpenes (51.5%). The biggest contribution to the oxygenated nor-diterpenes and diterpenes fraction was given by three components (**1**–**3**) that could not be identified by computer matching with MS libraries (laboratory made and commercial) and linear retention indices. Although quantitatively lower, the aromatic compounds were clearly present: 30 compounds accounting for 12.9% of the oil. Most of them were detected in trace amount (<0.1%).

The GC-MS result obtained in this study was significantly different from the previous published results on the leaf essential oil of *Dodonea viscosa* from the southern mountain region of Western Saudi Arabia [64]. The chemical constituents of this essential oil were reported to be four monoterpene hydrocarbons, two monoterpene alcohols (traces), five sesquiterpene hydrocarbons of which *cis*-caryophyllene was the major constituents, and three sesquiterpene alcohols (guaiol, β-eudesmol and a third unidentified compound).

### 2.2. Terpenoids Isolation and Identification

Repeated chromatographic purification (silica gel flash column and reverse phase-HPLC) of the essential oil of *Dodonea viscosa*, led to the isolation of pure compounds **1**–**3**.

A molecular formula C_19_H_24_O suggesting that **1** is an oxygenated nor-diterpene with eight degrees of unsaturation, was determined through HRESIMS (*m/z* [M + H]^+^ 269.1526, calcd 269.1905) and NMR data (Table 2). The HSQC and HMBC spectra (Table 2) revealed the presence of the following substructures: (a) a beta substituted furane ring (δC 110.0, 124.9, 137.3, 141.5), (b) a cycloheptatriene (CHT) moiety (δC 33.8, 123.2, 124.7, 128.0, 130.9, 131.1, 134.0) as well as (c) two methyl groups (δC 15.1, 22.0). Assembling the above substructures was mainly deduced from CH-correlations (Figure 2); that is, the furane was connected to the quaternary sp^3^ C-8 carbon atom carrying the singlet methyl C-19 via an ethylene bridge. Furthermore, C-8 was placed next to the CHT ring which was shown to be condensed to a methylcyclohexane ring. In addition, the furylethyl moiety was confirmed by EIMS fragments i.e., *m/z* 81 (furylmethylene fragment) as well as *m/z* 95 and *m/z* 173 (M-95) (furylethylene fragment). The above connectivities completed the gross structure of the molecule.

The relative stereochemistry of **1**, due to the twisted cyclohexene ring, making the NOE correlations ambiguous, could not be established. A computational study was therefore undertaken on its 8*R*, 9*S* and 8*R*, 9*R* isomers. Purposes of this density functional study were 1) to determine the relative configuration of dodovisate C through a comparison between calculated (for isomers 8*R*, 9*S* and 8*R*, 9*R*) and experimental ^13^C chemical shifts; 2) to elucidate the absolute stereochemistry of **1** via a comparison between experimental and calculated electronic circular dichroism spectra (ECD). As dodovisate C is a flexible molecule, its conformational analysis was therefore a prerequisite to the calculation of its ECD and NMR spectra. For the first task, geometries of the 8*R*, 9*S* and 8*R*, 9*R* isomers were optimized at the B3LYP/6-311+G(d,p) level. For each isomer, 12 conformations were optimized. The 12 figures of 8*R*, 9*S* (resp. 8*R*, 9*R*) conformers, their energies, dipole moments, relative energies and populations at 25 °C were gathered in Table 3 (resp. Table 4). Among them, there were two opposite boat conformations of the CHT ring, three different conformations of the furan ring (defined by the C13-C14-C15-C18 dihedral angle in Table 3 and Table 4) and two different twisted conformations of the cyclohexene ring leading to (2 × 3 × 2) = 12 conformers. All the conformers (**g–l** and **g’–l’**) with methyl group on C-9 in axial position were found to be more than 11.9 kJ/mol higher in energy than the most stable conformer of each isomer (a for 8*R*, 9*S* and f’ for 8*R*, 9*R*). Therefore, according to Boltzmann statistics, populations of **g–l** and **g’–l’** conformers were almost zero at 25 °C. Among the other 12 conformers a-f and a’-f’ with equatorial methyl group on C-9, conformers a-c and a’–c’ (resp. d–f, d’–f’) shared the same boat conformations of CHT ring.

^13^C NMR spectra of all conformers (with population > 0.1%) were calculated for each isomer at B3LYP/6-311+G(d,p) level. Then, calculated NMR chemical shifts were balanced thanks to Boltzmann statistics. Boltzmann-weighted chemical shifts versus experimental ones were plotted in Figure 3. The calculated ^13^C chemical shifts of 8*R*, 9*S* isomer clearly showed a better correlation to experiment than those of 8*R*, 9*R* isomer: the standard error of the fit s(8*R*, 9*S*) = 1.4 ppm was lower than s(8*R*, 9*R*) = 2.2 ppm; the Fisher F-statistic of isomer 8*R*, 9*S* was more than double of that of isomer 8*R*, 9*R*; standard errors on the slope, the intercept and the correlation coefficient were always lower for 8*R*, 9*S* isomer than for the 8*R*, 9*R* one’s. In a not surprising way, the calculated C-9*R* chemical shift had the largest deviation to experiment. In conclusion, the correlation of calculated versus experimental NMR chemical shifts clearly distinguished the 8S*, 9R* relative configuration from 8R*, 9R* one’s. A joint experimental and theoretical study using polarized light was carried out to assign the absolute configuration of dodovisate C. The experimental ECD spectrum of **1** is shown in Figure 4. In this spectrum, wavelengths of extreme values of Cotton effect were positive around 300 and 220 nm, negative around 260 nm. ECD spectra of the six populated conformers of 8*R*, 9*S* isomer were calculated at TD-B3LYP/6-311+G(d,p) level. Then, a Boltzmann-weighted calculated ECD spectrum was computed to allow comparison to experiment. In Figure 5, Boltzmann-weighted calculated ECD spectrum of 8*R*, 9*S* and 8*S*, 9*R* enantiomers were drawn. The comparison of calculated and experimental ECD spectra showed that the absolute configuration of dodovisate C isolated from the essential oil of *Dodonea viscosa* leaves was found to be 8*S*, 9*R*. For this enantiomer, calculated wavelengths of the positive (298 and 219 nm) and negative (259 nm) Cotton effects were in good agreement with experiment. The minimum energy DFT structure of this compound was shown in Figure 6. It had the same properties (energy, dipole moment, …) as conformer a in Table 3.

The second closely related isolated compound (**2**) displayed a *m/z* [M + H]^+^ peak at 327.1942 (calcd 327.1960) in the HRESIMS, consistent with a molecular formula of C_21_H_26_O_3_ and suggesting the occurrence of nine degrees of unsaturation. This formula was fully supported by NMR data (Table 2). Compound **2** was found to be identical in structure to **1** except for the carbomethoxyl moiety on the CHT ring. NMR data evidenced for **2**, a CHT ring like dodovisate C (**1**). But additionally, **2** carries a methyl ester at C-3, the required ninth double bond equivalent. The presence of the carbomethoxy group was suggested by the carbon resonances at δ 166.9 (s) and 51.9 (q), and it was confirmed by the HSQC correlation of the latter resonance to the three-proton resonance at δ 3.78 (s). Finally, compound **2** was found to be methyl dodovisate A, a modified clerodane previously isolated from same plant *Dodonea viscosa* by Nui et al. [60] As for dodovisate C (**1**), absence of NOE correlations prevented determination of the absolute configuration of methyl dodovisate A (**2**). Computational study was undertaken on methyl dodovisate A 8*R*, 9*S* and 8*R*, 9*R* isomers. Geometries of the 8*R*, 9*S* and 8*R*, 9*R* isomers were optimized exactly like dodovisate C. 12 conformations were optimized with methyl group on C-9 in equatorial position. The 12 figures of 8*R*, 9*S* (resp. 8*R*, 9*R*) conformers, their energies, dipole moments, relative energies, and populations at 25 °C were gathered in Table 5 and Table 6. The carbomethoxy group has two distinct positions. ^13^C-NMR spectra of all conformers (with population > 0.1%) were calculated for each isomer at B3LYP/6-311+G(d,p) level. Calculated NMR chemical shifts were weighted thanks to Boltzmann statistics. The calculated 13C chemical shifts of 8R, 9S isomer clearly showed a better correlation to experiment than those of 8R, 9R isomer: the standard error of the fit s(8R, 9S) = 1.7 ppm was lower than s(8R, 9R) = 2.1 ppm; the Fisher F-statistic of isomer 8R, 9S (F = 18342) is more important than that of isomer 8R, 9R (F = 11412); standard errors on the slope, the intercept and the correlation coefficient were presented in Table 7. Methyl dodovisate A (2) is suggested to be the isomer 8R*, 9S* or the isomer 8S*, 9R*.

Methyl iso-dodovisate A (**3**) was found to be an isomer of **2**, showing the same molecular formula C_21_H_26_O_3_. The ^1^H and ^13^C-NMR spectra of **3** were very similar to those of **2**, the only difference being the location of the three double bonds in the CHT ring. The structure of **3** (the 2,4,6-triene isomer) was achieved from the many HMBC correlations cross peaks (Table 2). The relative stereochemistry of the three chiral centers (C-1, C-8 and C-9) was deduced from the following NOE correlations, i.e., if H-1 is on one side of the bicyclic system the two methyl groups (Me-19 and Me-20) are on the opposite side. The key NOEs are from methyl 19 (δ 0.72) to H-2 (δ 6.31), H-6 (δ 6.13) and one of the H-10 (δ 1.54 m) protons establishing the twisting of the two rings (see Dreiding model). An NOE from methyl 19 to methyl 20 determined the latter two to be on the same side of the molecule. In this case the different twisting of the cyclohexane ring enabled to see the NOE between the two methyls (Figure 7). Computational study was performed on methyl iso-dodovisate A (**3**) for which, the carbon atoms 1, 8 and 9 are chiral. Geometries of the 1*S*, 8*R*, 9*R* (12 conformers with methyl group on C-9 in equatorial position); 1*S*, 8*R*, 9*S* (9 conformers with methyl group on C-9 in equatorial position); 1*S*, 8*S*, 9*R* (12 conformers with methyl group on C-9 in equatorial position) and 1*S*, 8*S*, 9*S* (7 conformers with methyl group on C-9 in equatorial position) isomers were optimized. The minimum energy DFT structure of this compound was shown in Table 8, Table 9, Table 10 and Table 11. There are two conformations of the CHT ring, one almost coplanar and the other one deeper in energy which represents a twisted cycle. Three different conformations of the furan ring defined by the C13-C14-C15-C18 dihedral angle exist, the carbomethoxy group has distinct orientations. The hexan 6 membered ring possesses two forms associated to the twisted CHT ring, the first one is in a boat-like form and the second one is in an envelope-like form. ^13^C-NMR spectra of all conformers (with population > 0.1%) were calculated for each isomer at B3LYP/6-311+G(d,p) level. Calculated NMR chemical shifts were weighted thanks to Boltzmann statistics. The calculated ^13^C chemical shifts of 1*S*, 8*R*, 9*R* isomer clearly showed a better correlation to experiment than the other three isomers: the standard error of the fit s(1*S*, 8*R*, 9*R*) = 1.7 ppm was the lowest; the Fisher F-statistic of isomer *1S,* 8*R*, 9*R* (F = 15132) is the more important factor; standard errors on the slope, the intercept and the correlation coefficient were presented in Table 7. Methyl iso-dodovisate A (**3**) is suggested to be the isomer 1*S**, 8*R**, 9*R** or the isomer 1*R**, 8*S**, 9*S**.

### 2.3. Discussion

No previous report on essential oils has described the presence of natural clerodanes whose decaline nucleus has been modified to a bicyclo[5.4.0]undecane, like dodovisate C (**1**), methyl dodovisate A (**2**) and methyl iso-dodovisate A (**3**). However, such bicyclic diterpenes have already been isolated from the organic extracts of a small number of species among which *Conyza scabrida* [65], several *Portulaca* species [66,67,68], *Baccharis linearis* [69], and *Dodonea viscosa* [63].

## 3. Materials and Methods

### 3.1. General Experimental Procedure

All solvents were HPLC or analytical grade. Silica gel 60 (particle size 63–200 µm, Machery-Nagel, Höerdt, France) was used for column chromatography. HPLC separations were performed on a LiChrospher RP-18 column (5 µm, 250 × 4 mm i.d., Merck, Darmstadt, Germany) with a Waters 600 pump coupled to a Waters 486 UV/Vis detector. Chromatographic data were collected and processed using the Empower software (Version 1, Waters, Milford, MA, USA). Optical rotation was determined on a Jasco P-1010 polarimeter (Tokyo, Japan) using a sodium lamp operating at 589 nm. ECD spectrum of **1** was measured at 25 °C in methanol at c = 4.85 × 10^−4^ mol·L^−1^ on a JASCO J-810 circular dichroism spectrometer (Tokyo, Japan). NMR spectra including NOESY, HMBC and HSQC experiments were recorded in CDCl_3_ on a Bruker Avance-600 spectrometer or on a Bruker Avance-500 spectrometer (Wissenbourg, France) operating at 600 and 500 MHz (^1^H) and 150 and 125 MHz (^13^C) respectively, with chemical shifts reported in ppm (δ) relative to TMS as internal standard. GC analyses were performed on a Varian Gas chromatograph Model CP-3800 (Walnut Creek, CA, USA), equipped with a flame ionization detection (FID) system (Walnut Creek, CA, USA) and a non-polar SPB-5 capillary column (60 m × 0.32 mm I.D., film thickness 0.25 µm, Bellefonte, USA). The oven temperature was programmed from 60 °C to 230 °C at 4 °C/min and then held isothermally at 230 °C for 40 min. Injector and detector temperatures were maintained at 250 °C and 300 °C, respectively. Nitrogen was used as the carrier gas at a flow rate of 0.7 mL/min. The samples diluted in CH_2_Cl_2_ were injected in splitless mode. GC-MS analyses (Palo Atto, CA, USA) were carried out using a Hewlett-Packard chromatograph type 6890 series equipped with a SPB-5 column (60 m × 0.32 mm i.d., film thickness 0.25 µm) and coupled to a HP 5972 mass selective detector. The MS detector was used in the EI mode with an ionization voltage of 70 eV over the *m/z* range 30–550. The carrier gas was Helium at a flow rate of 0.7 mL/min. The oven temperature was programmed from 60 °C to 230 °C at a rate of 4 °C/min, held for 40 min. The injector and the transfer line were both programmed to 250 °C. The samples diluted in CH_2_Cl_2_ were injected using a 20:1 split ratio.

### 3.2. Plant Material

The voucher number, the place and date of collection of *Dodonea viscosa* are given in Table 12. The specimen was identified by H. Thomas and deposited with the herbarium of Reunion Island (REU).

### 3.3. Essential oil Extraction

Fresh leaves were hydrodistilled in a Clevenger-type apparatus for 3 h. The oil was taken up in dichloromethane, dried over anhydrous sodium sulphate and kept at 4 °C. The extraction yield is reported in Table 12.

### 3.4. Isolation and Purification of Dodovisate C (1), methyl dodovisate A (2) and methyl iso-dodovisate A (3)

The essential oil (988.0 mg) was chromatographed on a silica gel flash column eluted successively with iso-hexane (150 mL) and EtOAc (150 mL). The *iso*-hexane fraction (33.9 mg) containing dodovisate C (**1**), was then further submitted to RP-HPLC purification using a gradient of elution (20% CH_3_CN-H_2_O to 100% CH_3_CN over 30 min, then maintained at 100% CH_3_CN for 10 min) and a flow rate of 1.0 mL/min, to afford 5.2 mg of **1** (colorless, amorphous solid).

The essential oil (86.0 mg) was chromatographed on a silica gel flash column eluted successively with *iso*-hexane (150 mL) and EtOAc (150 mL). The EtOAc fraction (22.9 mg) containing methyl dodovisate A (**2**) and methyl iso-dodovisate A (**3**), was then further submitted to repeated column chromatography over flash silica gel with *n*-hexane–EtOAc gradient to afford 13.9 mg of **2** and 1.3 mg of **3** (light yellowish oils with a strong aromatic odours). The fractions were monitored by GC-MS.

### 3.5. Identification and Quantification

The identification of individual components was based on: (a) comparison of calculated linear retention indices (LRIs), on apolar columns with those of literature data [70,71,72]; (b) computer matching with commercial mass spectral libraries (NIST and Wiley) and comparison of mass spectra with those of our laboratory-built library or literature data [70,71]; (c) 1D- (^1^H, ^13^C, APT) and 2D-(HSQC, HMBC, NOESY) NMR spectra only for dodovisate C (**1**), methyl dodovisate A (**2**) and methyl iso-dodovisate A (**3**) after isolation and purification. The quantification of the components was performed on the basis of their GC peak areas on the SPB-5 column without FID response factor correction.

Dodovisate C (*8-(2-furanethyl)-8,9-dimethylbicyclo[5.4.0]undeca-1,3,5-triene*, **1**): Colorless amorphous solid: ${\left[\alpha \right]}_{D}^{20}$ +4.85 (*c* 0.21, 23 °C, CH_2_Cl_2_); EI-MS 70 eV, *m*/*z* (rel. int.): 268 [M^+^] (17), 253(3), 240(8), 173(100), 159(13), 145(24), 131(74), 117(66), 104(37), 91(35), 81(26), 69(9), 53(12), 41(14); NMR spectral data: see Table 2.

Methyl dodovisate A (*3-acetoxy-8-(2-furanethyl)-8,9-dimethylbicyclo[5.4.0]undeca-1,3,5-triene*, **2**): Yellowish oil: EI-MS 70 eV *m/z* (rel. int.): 326 [M]^+^ (18), 311 [M − CH_3_]^+^ (5), 295 [M − CH_3_ −H_2_O]^+^ (4), 267 (7), 231 (100), 217 (9), 199 (28), 189 (16), 175 (28), 163 (35), 157 (21), 149 (18), 129 (27), 128 (27), 115(19), 105 (9), 95 (19), 81 (25), 69 (17), 59 (9), 41 (9); NMR spectral data: see Table 2.

Methyl iso-dodovisate A (*3-acetoxy-8-(2-furanethyl)-8,9-dimethylbicyclo[5.4.0]undeca-2,4,6-triene*, **3**): Yellowish oil: EI-MS 70 eV *m/z* (rel. int.): 326 [M]^+^ (18), 311 [M − CH_3_]^+^ (5), 295 [M − CH_3_ −H_2_O]^+^ (4), 267 (7), 231 (100), 217 (9), 199 (28), 189 (16), 175 (28), 163 (35), 157 (21), 149 (18), 129 (27), 128 (27), 115(19), 105 (9), 95 (19), 81 (25), 69 (17), 59 (9), 41 (9); NMR spectral data: see Table 2.

The compounds NMR and MS data of compounds **1** and **3** can be found in the Appendix A.

### 3.6. Computational Details

All DFT calculations were carried out using the GAUSSIAN 09 program [73] using the hybrid B3LYP exchange-correlation functional [74,75] and the 6-311+G(d,p) basis set. Tight convergence criteria were used for geometry optimization. All stationary points were confirmed as true minima via vibrational frequency calculations. Frequencies calculated in the harmonic approximation were multiplied with a factor set to 0.98.

NMR shielding tensors were computed with the Gauge-Independent Atomic Orbital (GIAO) method [76]. The calculated nuclear isotropic magnetic shieldings (IMS) were converted for carbons C–i into chemical shifts by referencing to tetramethylsilane (TMS) IMS calculated at the same level of theory (IMS_TMS_ = 184.1416 ppm): δ_corr_ = IMS_TMS_ − IMS_C–i_.

Electronic excitation energies and rotational strengths for all conformations have been calculated using the TDDFT methodogy of Gaussian 09 for the 150 lowest singlet vertical excitation energies at ground-state equilibrium geometries. Thence ECD spectra were obtained assuming the Condon approximation and the computed velocity rotatory strengths are transformed into units of **Δ**ε and superimposed with Gaussian functions centered at the respective wavenumbers of the electronic transitions. An exponential half-width at 1/e peak height **Δ**σ = 0.4 eV was used for each gaussian curve [77].

Molecular structures were done with Molden software [78].

## Figures and Tables

**Figure 1 molecules-25-00850-f001:**
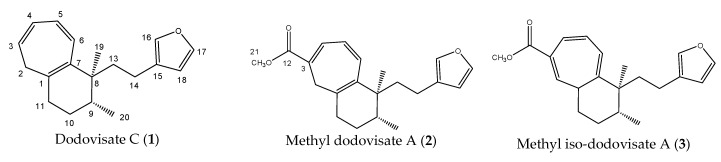
Structure of the three cyclopropylclerodanes isolated from *Dodonea viscosa*.

**Figure 2 molecules-25-00850-f002:**
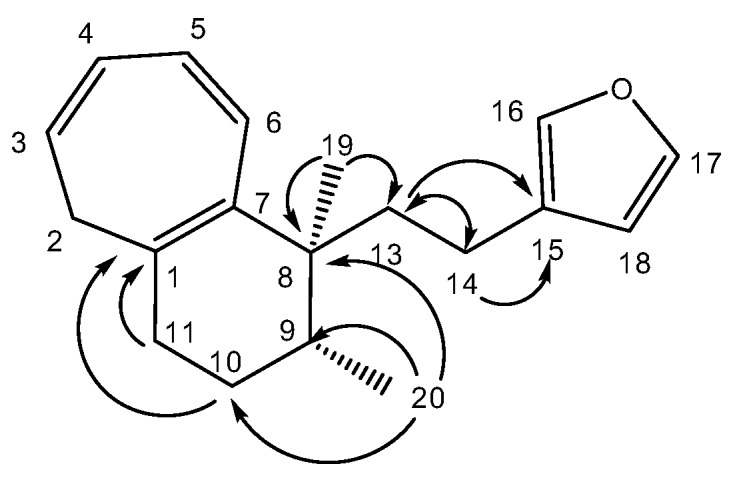
Selected HMBC correlations for compound (**1**).

**Figure 3 molecules-25-00850-f003:**
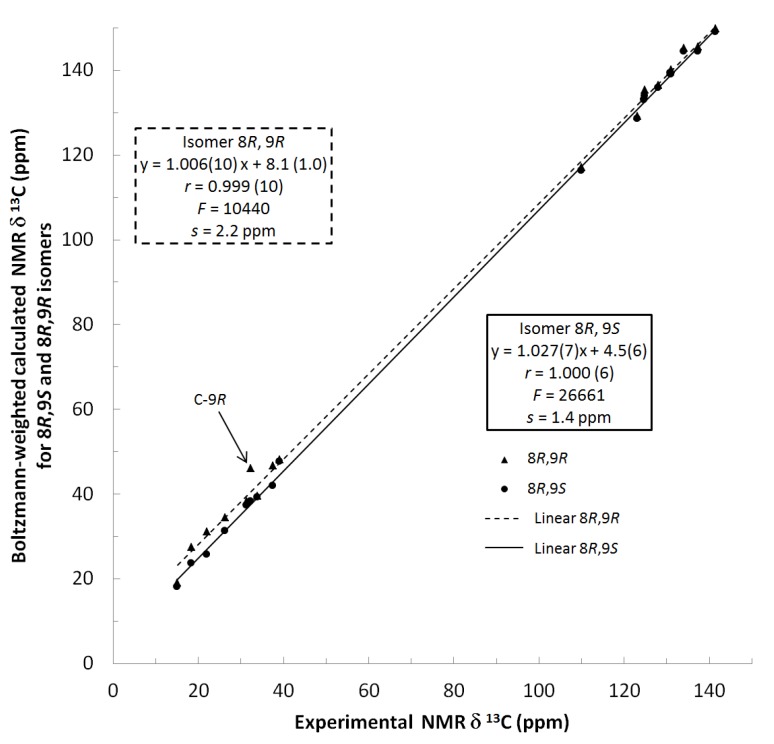
Plot of Boltzmann-weighted calculated NMR δ^13^C of 8*R*, 9*S* and 8*R*, 9*R* isomers *versus* experimental NMR δ^13^C of dodovisate C (**1**). Statistics for the regression of calculated *versus* experimental chemical shifts for both isomers. The slope, the intercept, the correlation coefficient (r) are followed by their standard error on the last digit(s) in brackets. F is the Fisher F-statistic and s the standard error of the fit.

**Figure 4 molecules-25-00850-f004:**
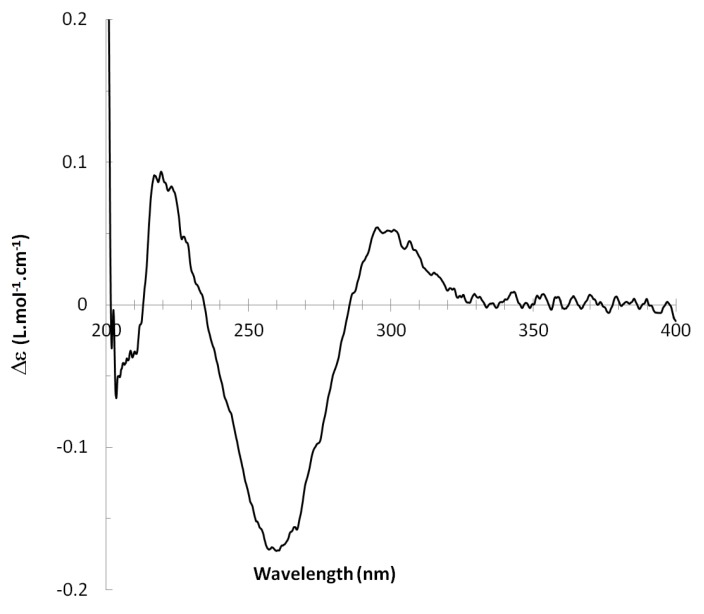
Experimental ECD spectrum of dodovisate C (**1**).

**Figure 5 molecules-25-00850-f005:**
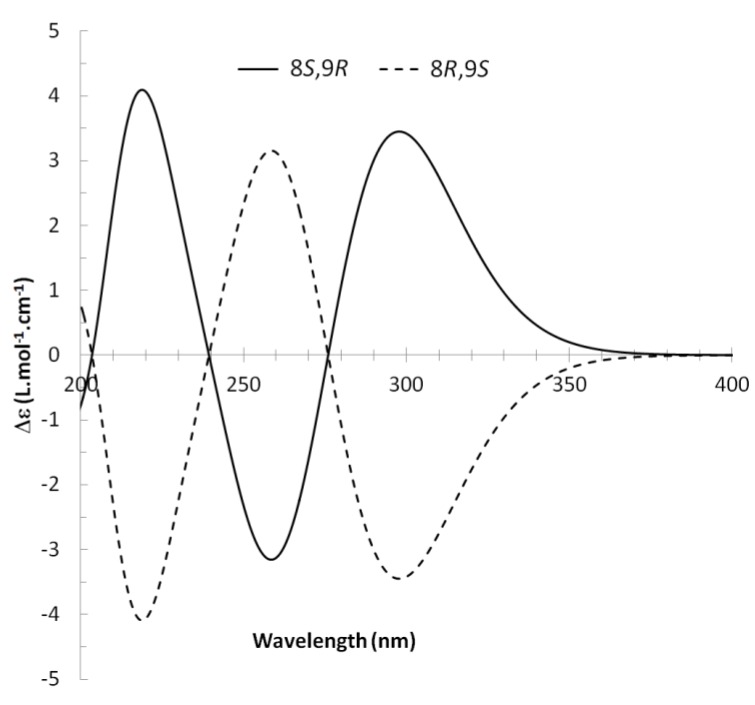
Boltzmann-weighted calculated ECD spectra of 8*R*, 9*S* and 8*S*, 9*R* enantiomers of dodovisate C (**1**).

**Figure 6 molecules-25-00850-f006:**
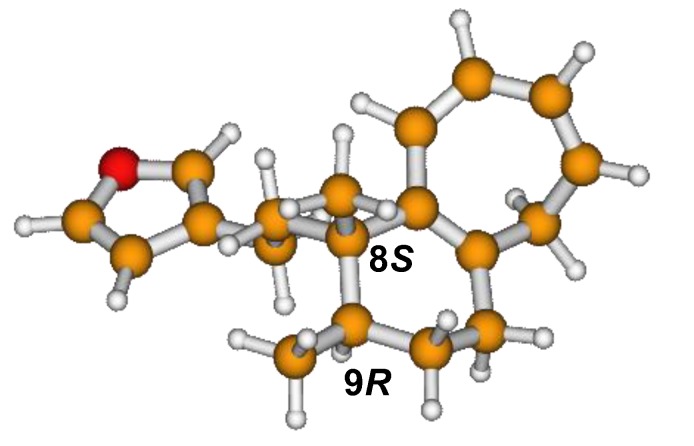
Minimum B3LYP/6-311+G(d,p) energy structure of 8-(2-furanethyl)-8*S*,9*R*-dimethylbicyclo[5.4.0]undeca-1,3,5-triene or dodovisate C (**1**) isolated from essential oil of *Dodonea viscosa* leaves.

**Figure 7 molecules-25-00850-f007:**
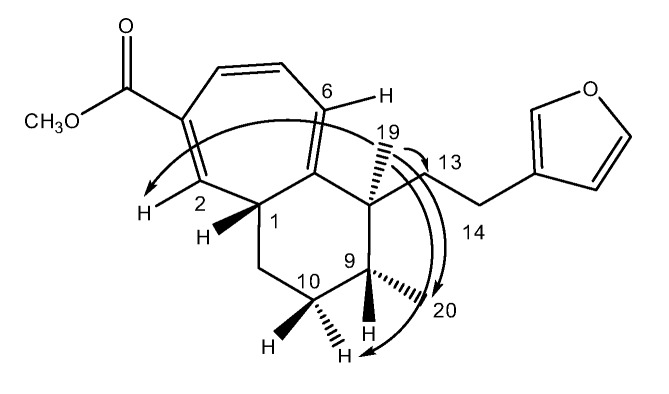
Selected NOE correlations for compound **3**.

**Table 1 molecules-25-00850-t001:** Chemical composition of the essential oil of leaves from *Dodonea viscosa.*

	**Compounds ^a^**	**RI ^b^**	**% ^c^**
Hydrocarbons			
	*n*-heneicosane	2097	tr
	*n*-pentacosane	2495	tr
	total		tr
Alcohols			
	hex-3-enol (*Z/E* configuration n.i.)	853	0.8
	hex-2-enol (*Z/E* configuration n.i.)	861	tr
	*n*-hexanol	864	0.8
	oct-1-en-3-ol	978	*tr*
	*n*-octanol	1069	m (1.3)
	*n*-decanol	1271	tr
	total		2.9
Ketones			
	isophorone	1125	tr
	1-phenylbut-2-enone	1312	0.3
	(*Z*)-jasmone	1401	tr
	geranylacetone	1452	0.2
	(*E*)-β-ionone	1490	0.2
	6,10,14-trimethylpentadecan-2-one	1843	1.1
	total		1.8
Carboxylic acids			
	isovaleric acid	826	tr
	2-methylbutanoic acid	837	tr
	hexanoic acid	973	0.3
	benzoic acid	1159	tr
	octanoic acid	1167	tr
	dodecanoic acid	1559	0.2
	tetradecanoic acid	1759	0.7
	hexadecanoic acid	1963	2.4
	total		3.6
Esters			
	isopentyl isovalerate	1104	1.5
	*n*-hexyl butanoate	1190	*tr*
	*n*-hexyl 2-methylbutanoate	1236	0.4
	*n*-hexyl 3-methylbutanoate	1239	2.3
	total		4.2
Oxygenated monoterpenes			
	*trans*-linalool oxide (furanoid)	1074	0.1
	*cis*-linalool oxide (furanoid)	1091	tr
	linalool	1100	1.0
	terpinen-4-ol	1182	0.4
	α-terpineol	1195	0.3
	nerol	1230	0.2
	geraniol	1255	tr
	total		2.0
Sesquiterpene hydrocarbons			
	aromadendrene	1449	0.2
	(*E*,*E*)-α-farnesene	1508	tr
	δ-cadinene	1530	tr
	α-calacorene	1551	tr
	total		0.2
Oxygenated sesquiterpenes			
	(*E*)-nerolidol	1565	0.2
	(*Z*)-dihydro-apofarnesol	1579	tr
	spathulenol	1588	0.5
	globulol	1595	0.5
	viridiflorol	1604	0.2
	epi-α-muurolol	1658	tr
	α-cadinol	1664	0.2
	(2*E*,6*Z*)-farnesol	1744	tr
	total		1.6
Oxygenated diterpenes			
	isophytol	1947	tr
	(1)	2036	35.0
	phytol	2111	0.5
	(2) + (3)	2420	m (16.0)
	hardwickiic acid, methyl ester	2431	m (16.0)
	total		51.5
Aromatic compounds			
	benzaldehyde	963	0.1
	benzyl alcohol	1035	1.8
	benzene acetaldehyde	1045	0.1
	*o*-cresol	1054	0.7
	acetophenone	1068	m (1.3)
	phenylethyl alcohol	1116	tr
	methyl salicylate	1199	0.5
	4-vinyl phenol	1218	0.1
	chavicol	1254	tr
	*p*-anisaldehyde dimethyl acetal	1256	tr
	*p*-ethylacetophenone	1285	tr
	1-phenylbut-2-en-1-one	1312	0.3
	2-methoxy-4-vinylphenol	1317	0.5
	benzyl butanoate	1346	tr
	eugenol	1360	0.9
	methyl *p*-anisate	1378	tr
	benzyl isovalerate	1395	2.1
	vanillin	1401	tr
	methyl eugenol	1403	0.4
	2-methylbutyl benzoate	1440	0.2
	ethyl vanillin	1461	tr
	phenylethyl 3-methylbutanoate	1493	0.5
	asaricin	1500	tr
	isopentyl salicylate	1539	0.1
	(3*Z*)-hexenyl benzoate	1574	0.8
	*n*-hexyl benzoate	1580	1.0
	(3Z)-hexenyl salicylate	1673	0.2
	*n*-hexyl salicylate	1681	0.2
	benzyl benzoate	1771	0.6
	benzyl salicylate	1877	0.5
	total		12.9
Others			
	7-methyl-1,6-dioxaspiro [4,5] decane (stereoisomer n.i.)	1058	1.1
	vitispirane	1286	tr
	total		1.1
Total identified			80.5

^a^ n.i.: Non identified; ^b^ LRI: Linear retention index calculated on non-polar (SPB-5) column; ^c^ Relative percentage based on the peak area from the GC-MS analysis; tr: trace (<0.1%).

**Table 2 molecules-25-00850-t002:** 1D and 2D NMR spectroscopic data for dodovisate C (**1**), methyl dodovisate A (**2**), methyl iso-dodovisate A (**3**).

Position	Dodovisate C (1)			Methyl Dodovisate A (2)			Methyl Iso-Dodovisate A (3)		
	δ ^13^C (150 MHz)	δ ^1^H (600 MHz)	HMBC (H→C)	δ ^13^C (500 MHz)	δ ^1^H (125 MHz)	HMBC (H→C)	δ ^13^C (500 MHz)	δ ^1^H (125 MHz)	HMBC (H→C)
1	130.9 s			135.5 s			38.3 d	1.52 m ^a^	2, 4, 6, 7, 10, 11
2	33.8 t	2.28 dd (12.3, 7.0) 2.07 dd (12.3, 6.8)	1, 3, 4, 7, 11	33.3 t	2.30 m 2.71 bd (11.5)	1, 3, 4, 7	135.0 d	6.31 d (3.7)	1, 4, 5w, 11
3	123.2 d	5.49 ddd (8.9, 7.0, 6.8)	1, 2, 5	123.7 s			126.8 s		
4	124.7 d	5.98 dd (8.9, 5.1)	2, 6	132.0 d	7.10 d (5.6)	2, 3, 5, 6	125.9 d	7.06 d (11.1)	2, 6
5	128.0 d	6.40 dd (11.4, 5.1)	3, 4, 7	127.6 d	6.58 dd (11.6, 5.6)	1, 3, 4	131.5 d	6.75 dd (11.1, 5.7)	2 ^w^, 4, 6, 7
6	131.1 d	6.64 d (11.4)	1, 4, 5, 7, 8	136.6 d	6.97 d (11.6)	3, 4, 5, 7, 8	119.5 d	6.13 d (5.7)	4
7	134.0 s			134.5 s			143.5 s		
8	39.1 s			40.4 s			42.3 s		
9	32.3 d	1.69 m	7, 8, 19	33.2 d	1.78 m	19	34.9 d	1.89 m ^a^	19
10	26.2 t	1.40 m	1, 8, 9	26.9 t	1.45 m 1.55 m	1, 8, 9	26.3 t	1.54 m ^a ^ 1.73 m (dquin like)	1 ^w^, 8, 11
11	31.4 t	2.16 m 2.34 m	1, 7, 10	31.8 t	2.28 m 2.43 m	1, 7, 10	28.4 t	1.78 m2.02 m	1 ^w^, 9, 10
12				166.9 s			167.6 s		
13	37.5 t	1.69 m	7, 8, 9, 14, 15	38.3 t	1.80 m	8, 9, 14	37.0 t	1.83 m ^a^, 1.89 m^a^	
14	18.4 t	1.88 m 2.16 m	13, 15, 16, 18	19.5 t	1.97 m 2.23 m	13, 15, 16, 18	19.6 t	2.53 dbrt (3.8, 12.2) 2.43 dbrt (5.2, 12.2)	9 ^w^, 13, 15, 16, 18
15	124.9 s			125.6 s			125.6 s		
16	110.0 d	6.18 s	15, 17, 18	110.9 d	6.25 s	15, 17, 18	110.9 d	6.34 brs	15 ^w^, 17, 18
17	141.5 d	7.20 s	15, 16, 18	142.6 d	7.34 s	15, 16, 18	142.8 d	7.34 brs	15, 16
18	137.3 d	7.10 s	15, 16, 17	138.4 d	7.18 s	15, 16, 17	138.4 d	7.25 brs	15, 16, 17
19	22.0 q	0.85 s	7, 8, 9, 13	23.3 q	0.88 s	8, 13	30.0 q	0.72 s	7, 8, 9, 13, 14 ^w^
20	15.1 q	0.82 d (3.6)	8, 9, 10	15.9 q	0.83 d (6.8)	8, 9, 10	16.3 q	0.81 d (7)	8, 9, 10
21				51.9 q	3.78 s	12	51.8 q	3.75 s	12

^a^ Overlapping signals; ^w^ weak signal.

**Table 3 molecules-25-00850-t003:** Calculated B3LYP/6-311+G(d,p) Energies ^a^, Dipole moments (Debye), relatives Energies^a^, (C13-C14-C15-C18) dihedral angle, Equilibrium population at 25°C and figures of conformations **a-l** of the 8*R*, 9*S* isomer of dodovisate C (**1**).

**Conformer**	**a**	**b**	**c**	**d**	**e**	**f**
Figure	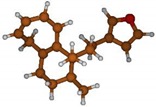	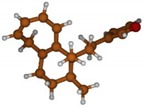	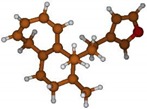	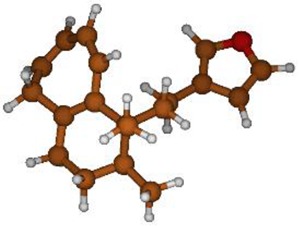	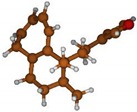	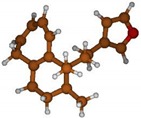
E (ua)	−813.474252	−813.473483	−813.474156	−813.473265	−813.472512	−813.473392
Dipole moment	1.83	1.77	1.47	1.85	1.81	1.57
Dihedral angle	−111.4°	1.2°	110.3°	−110.7°	1.3°	110.2°
ΔE (kJ/mol)	0.00	2.02	0.25	2.59	4.57	2.26
Population (%)	30.7	13.6	27.7	10.8	4.9	12.3
Conformer	**g**	**h**	**i**	**j**	**k**	**l**
Figure	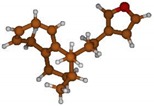	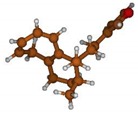	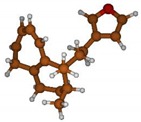	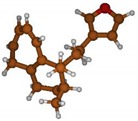	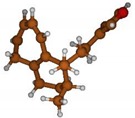	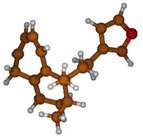
E (ua)	−813.467509	−813.466609	−813.467563	−813.468819	−813.467850	−813.469001
Dipole moment	1.65	1.66	1.34	1.71	1.72	1.43
Dihedral angle	−111.0°	5.4	109.1	−109.8	4.3	110.0
ΔE (kJ/mol)	17.70	20.07	17.56	14.26	16.81	13.79
Population (%)	0.0	0.0	0.0	0.0	0.0	0.0

^a^ Include zero-point vibration correction.

**Table 4 molecules-25-00850-t004:** Calculated B3LYP/6-311+G(d,p) Energies ^a^, Dipole moments (Debye), relatives Energies^a^, (C13-C14-C15-C18) dihedral angle, equilibrium population at 25°C and figures of conformations a-l of the 8*R*, 9*R* isomer of dodovisate C (**1**).

**Conformer**	**a’**	**b’**	**c’**	**d’**	**e’**	**f’**
Figure	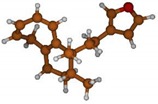	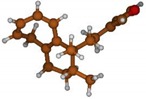	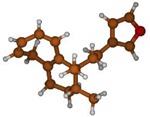	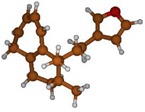	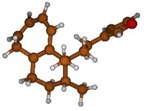	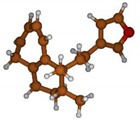
E (ua)	−813.463138	−813.462352	−813.463090	−813.464746	−813.463918	−813.464838
Dipole moment	1.76	1.74	1.46	1.82	1.81	1.56
Dihedral angle	−111.4°	4.8°	109.4°	−110.9°	3.5°	110.3°
ΔE (kJ/mol)	4.46	6.53	4.59	0.24	2.42	0.00
Population (%)	6.1	2.7	5.8	33.6	14.0	37.1
Conformer	**g’**	**h’**	**i’**	**j’**	**k’**	**l’**
Figure	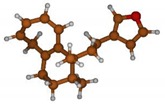	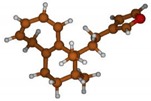	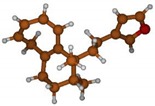	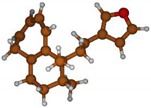	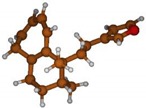	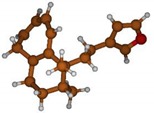
E (ua)	−813.460304	−813.459717	−813.460261	−813.458557	−813.457959	−813.4585678
Dipole moment	1.95	1.78	1.60	1.97	1.79	1.64
Dihedral angle	−112.7°	1.8°	113.4°	−111.6°	0.9°	112.7
ΔE (kJ/mol)	11.91	13.45	12.02	16.49	18.06	16.46
Population (%)	0.3	0.0	0.3	0.0	0.0	0.0

^a^ Include zero-point vibration correction.

**Table 5 molecules-25-00850-t005:** Calculated B3LYP/6-311+G(d,p) Energies ^a^, Dipole moments (Debye), relatives Energies^a^, (C_13_-C_14_-C_15_-C_18_) dihedral angle, equilibrium population at 25°C and figures of conformations 1-12 of the 8*S*, 9*R* isomer of methyl dodovisate A (**2**).

**Conformer**	**5**	**10**	**4**	**11**	**12**	**8**
Figure	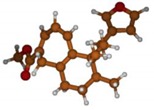	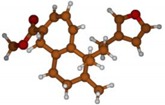	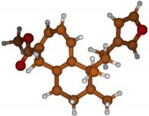	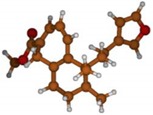	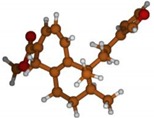	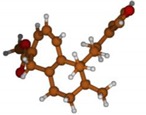
E (ua)	−1041.381622	−1041.380520	−1041.381631	−1041.380467	−1041.379800	−1041.380902
Dipole moment	1.08	3.62	0.87	3.12	3.12	0.35
Dihedral angle	−111.6°	−112.5°	110.5°	109.7°	3.6°	2.4°
ΔE (kJ/mol)	3.15	6.04	3.12	6.18	7.93	5.04
Population (%)	7.2	2.2	7.2	2.1	1.1	3.3
Conformer	**6**	**1**	**7**	**2**	**3**	**9**
Figure	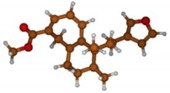	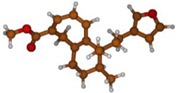	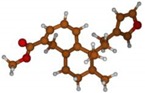	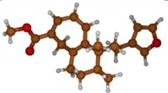	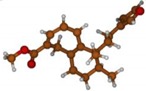	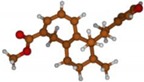
E (ua)	−1041.381577	−1041.382821	−1041.381499	−1041.382724	−1041.382110	−1041.380875
Dipole moment	2.92	1.61	2.05	1.73	0.98	2.70
Dihedral angle	−111.8°	−111.8	110.4	110.4	1.3	1.2
ΔE (kJ/mol)	3.27	0.00	3.47	0.25	1.87	5.11
Population (%)	6.8	25.5	6.3	23.0	12.0	3.3

^a^ Include zero-point vibration correction.

**Table 6 molecules-25-00850-t006:** Calculated B3LYP/6-311+G(d,p) Energies ^a^, Dipole moments (Debye), relatives Energies^a^, (C_13_-C_14_-C_15_-C_18_) dihedral angle, equilibrium population at 25°C and figures of conformations 1-12 of the 8*R*, 9*R* isomer of methyl dodovisate A (**2**).

**Conformer**	**4**	**2**	**5**	**1**	**3**	**8**
Figure	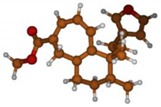	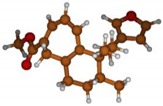	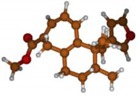	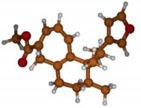	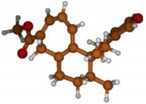	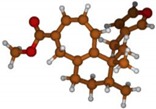
E (ua)	−1041.379759	−1041.380966	−1041.379717	−1041.381006	−1041.380144	−1041.378913
Dipole moment	3.65	1.13	3.12	0.98	0.28	3.20
dihedral angle	−112.4°	−111.7	109.5°	110.6°	3.7°	6.0°
ΔE (kJ/mol)	3.27	0.11	3.39	0.00	2.26	5.49
Population (%)	7.8	28.0	7.5	29.2	11.7	3.2
Conformer	**11**	**7**	**10**	**6**	**9**	**12**
Figure	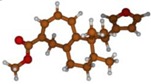	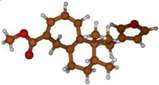	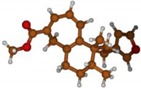	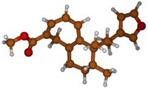	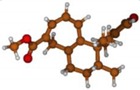	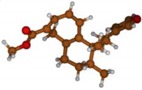
E (ua)	−1041.378027	−1041.379098	−1041.378051	−1041.379135	−1041.378310	−1041.377274
Dipole moment	2.73	1.96	1.93	1.86	1.22	2.61
dihedral angle	−111.8°	−111.7°	109.6°	109.5°	4.9°	4.9°
ΔE (kJ/mol)	7.82	5.01	7.76	4.91	7.08	9.80
Population (%)	1.2	3.9	1.3	4.0	1.7	0.5

^a^ Include zero-point vibration correction.

**Table 7 molecules-25-00850-t007:** Showing linear regressions of theoretical chemical shifts *versus* experimental values for isomers of dodovisate C (**1**), methyl dodovisate A (**2**) and methyl iso-dodovisate A (**3**) standard errors on the slope, the intercept and the correlation coefficient, Fischer-F statistic factor and standard error of the fit in ppm.

Compound	Isomers	Slope	Intercept	Coefficient Correlation	Fischer-F Statistic	s in ppm
Dodovisate C (**1**)	8*R*,9*S*	1.027(7)	4.5(6)	1.000(6)	26661	1.4
	8*R*,9*R*	1.006(10)	8.1(1.0)	0.999(10)	10440	2.2
Methyl dodovisate A (**2**)	8*R*,9*S*	0.9685(7)	4.7(7)	0.999(7)	18342	1.7
	8*R*,9*R*	0.9453(9)	6.2(9)	0.999(9)	11412	2.1
Methyl iso-dodovisate A (**3**)	1*S*,8*R*,9*R*	0.9405(7)	7.0(8)	0.999(8)	15132	1.8
	1*S*,8*R*,9*S*	0.9610(10)	4.3(1.0)	0.999(10)	10007	2.3
	1*S*,8*S*,9*R*	0.9479(10)	6.2(1.9)	0.996(20)	2593	4.4
	1*S*,8*S*,9*S*	0.9613(13)	4.5(1.3)	0.998(13)	5556	3.1

**Table 8 molecules-25-00850-t008:** Calculated B3LYP/6-311+G(d,p) Energies ^a^, Dipole moments (Debye), relatives Energies^a^, (C13-C14-C15-C18) dihedral angle, equilibrium population at 25°C and figures of conformations 1-12 of the 1*S*,8*R*,9*R* isomer of methyl iso-dodovisate A (**3**).

**Conformer**	**7**	**10**	**9**	**12**	**11**	**5**
Figure	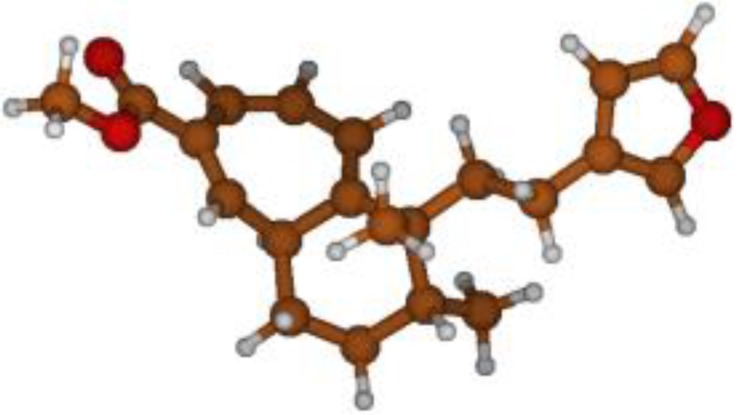	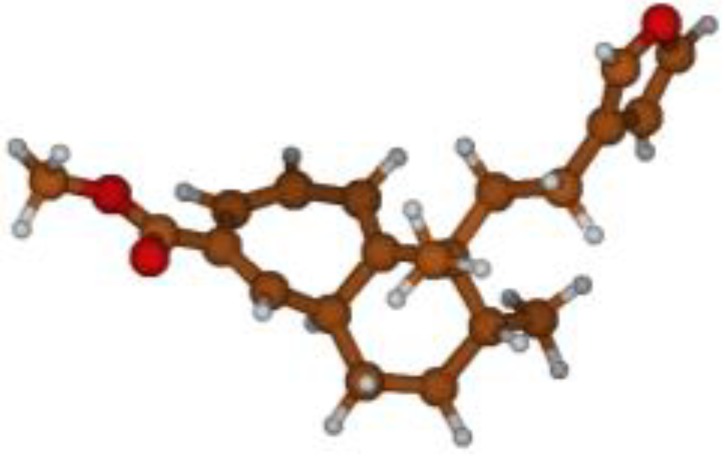	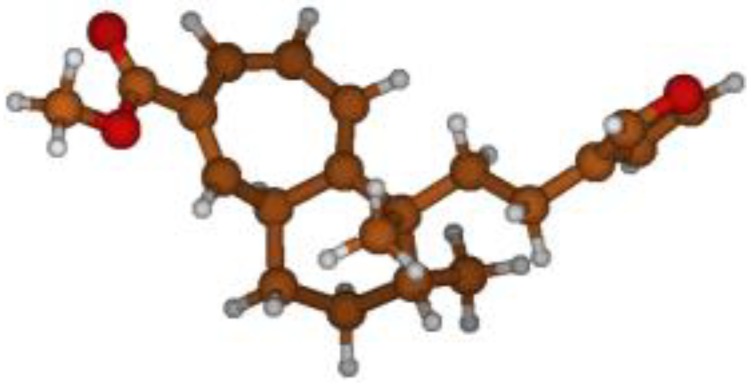	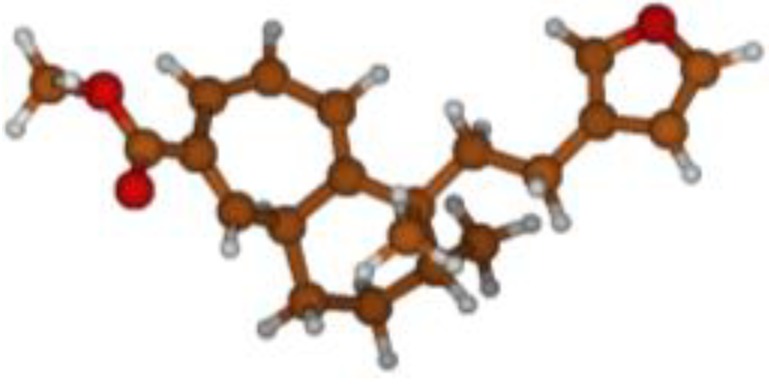	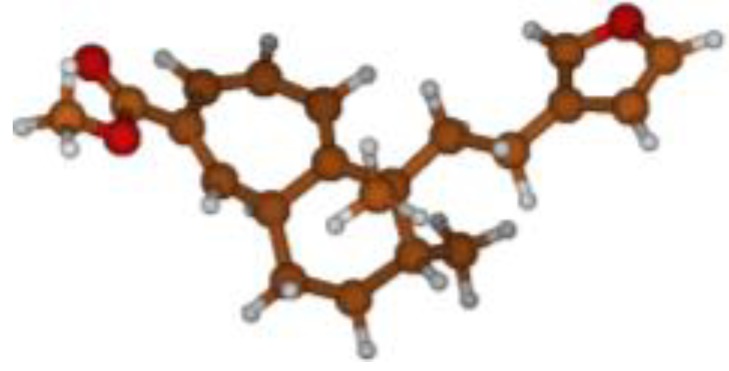	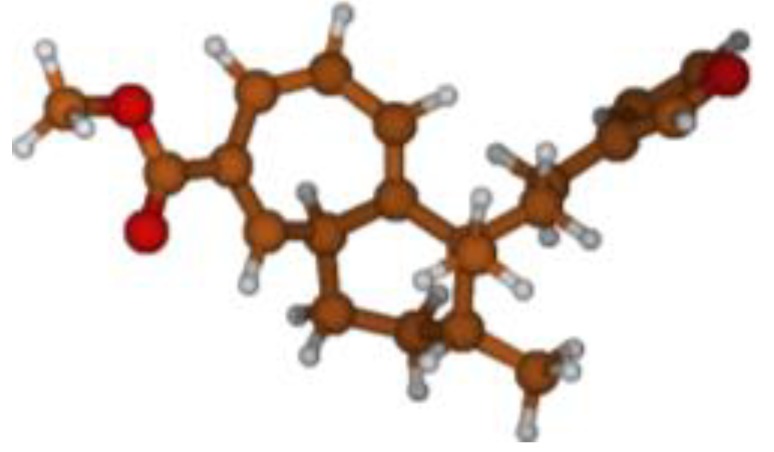
E (ua)	−1041.365849	−1041.365746	−1041.365780	−1041.364999	−1041.365036	−1041.366448
Dipole moment	2.90	1.61	3.20	1.11	3.72	1.01
dihedral angle	−114.6°	112.0°	112.2°	1.0°	1.0°	2.7°
ΔE (kJ/mol)	3.44	3.71	3.62	5.68	5.58	1.87
Population (%)	4.2	3.7	3.9	1.7	1.7	7.9
Conformer	**6**	**8**	**2**	**1**	**4**	**3**
Figure	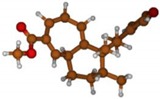	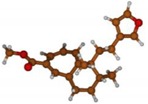	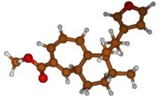	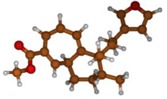	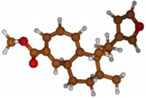	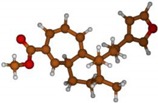
E (ua)	−1041.366428	−1041.365784	−1041.367137	−1041.367160	−1041.367124	−1041.367129
Dipole moment	3,11	0,82	0,75	3,58	0,70	2,74
dihedral angle	2.8°	−114.4°	−112.9°	−112.9°	110.8°	110.7°
ΔE (kJ/mol)	1.92	3.61	0.06	0.00	0.10	0.08
Population (%)	7.7	3.9	16.3	16.7	16.1	16.2

^a^ include zero-point vibration correction.

**Table 9 molecules-25-00850-t009:** Calculated B3LYP/6-311+G(d,p) Energies ^a^, Dipole moments (Debye), relatives Energies^a^, (C13-C14-C15-C18) dihedral angle, equilibrium population at 25°C and figures of conformations 1-9 of the 1*S*, 8*R*, 9*S* isomer of methyl iso-dodovisate A (**3**).

**Conformer**	**3**	**1**	**4**	**2**	**5**	**6**
Figure	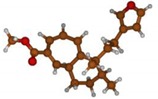	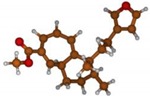	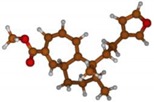	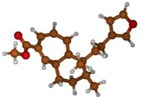	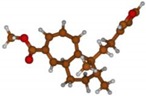	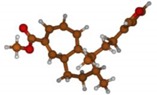
E (ua)	−1041.365913	−1041.365955	−1041.365867	−1041.365937	−1041.365388	−1041.365382
Dipole moment	1.21	3.73	0.97	2.88	1.73	3.31
dihedral angle	−116.4°	−116.1°	112.4	112.2°	1.7°	2.0°
ΔE (kJ/mol)	0.11	0.00	0.3	0.05	1.49	1.50
Population (%)	17.9	18.7	17.0	18.3	10.2	10.2
Conformer	**8**	**9**	**7**
Figure	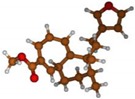	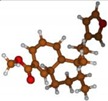	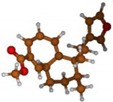
E (ua)	−1041.364092	−1041.364040	−1041.364102
Dipole moment	0.77	0.76	2.71
dihedral angle	−112.5°	110.4	110.4°
ΔE (kJ/mol)	4.89	5.03	4.86
Population (%)	2.6	2.5	2.6

^a^ include zero-point vibration correction.

**Table 10 molecules-25-00850-t010:** Calculated B3LYP/6-311+G(d,p) Energies ^a^, Dipole moments (Debye), relatives Energies^a^, (C13-C14-C15-C18) dihedral angle, equilibrium population at 25°C and figures of conformations 1-12 of the 1*S*, 8*S*, 9*R* isomer of methyl iso-dodovisate A (**3**).

**Conformer**	**1**	**2**	**5**	**10**	**3**	**4**
Figure	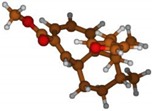	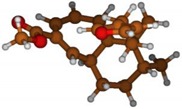	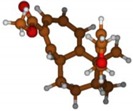	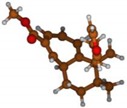	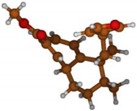	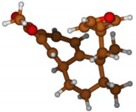
E (ua)	−1041.369364	−1041.368947	−1041.367744	−1041.367199	−1041.368890	−1041.368444
Dipole moment	2.51	3.23	2.36	2.49	2.51	2.60
dihedral angle	−105.1°	−105.8°	2.8°	3.9°	92.7°	97.5
ΔE (kJ/mol)	0.00	1.09	4.25	5.68	1.24	2.42
Population (%)	28.7	18.	5.2	2.9	17.4	10.8
Conformer	**9**	**8**	**12**	**7**	**6**	**11**
Figure	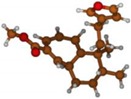	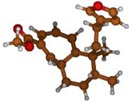	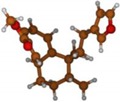	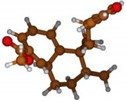	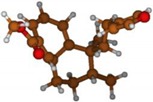	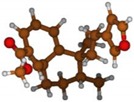
E (ua)	−1041.367202	−1041.367402	−1041.366173	−1041.367417	−1041.367530	−1041.366354
Dipole moment	2.40	3.50	2.50	2.92	2.12	2.69
dihedral angle	−107.4°	−109.8°	−2.1°	109.9°	111.0°	0.6°
ΔE (kJ/mol)	5.68	5.15	8.38	5.11	4.82	7.90
Population (%)	2.9	3.6	1.0	3.7	4.1	1.2

^a^ include zero−point vibration correction.

**Table 11 molecules-25-00850-t011:** Calculated B3LYP/6-311+G(d,p) Energies ^a^, Dipole moments (Debye), relatives Energies^a^, (C13-C14-C15-C18) dihedral angle, equilibrium population at 25°C and figures of conformations 1-7 of the 1*S*, 8*S*, 9*S* isomer of methyl iso-dodovisate A (**3**).

**Conformer**	**1**	**3**	**5**	**6**	**2**	**4**
Figure	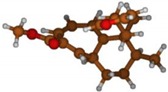	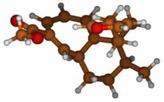	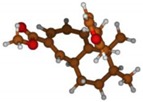	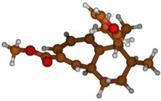	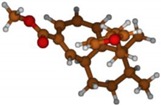	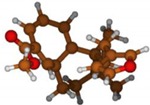
E (ua)	−1041.368995	−1041.368611	−1041.367491	−1041.366883	−1041.368632	−1041.368189
Dipole moment	2,58	3.24	2.38	2.56	2.57	2.66
dihedral angle	−104.0°	−105.3°	2.4°	3.7°	94.4°	98.6°
ΔE (kJ/mol)	0.00	1.01	3.95	5.55	0.95	2.12
Population (%)	32.4	21.6	6.6	3.5	22	13.8
Conformer	**7**
Figure	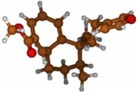
E (ua)	−1041.362938
Dipole moment	1.89
dihedral angle	110.9°
ΔE (kJ/mol)	15.90
Population (%)	0.1

^a^ include zero-point vibration correction.

**Table 12 molecules-25-00850-t012:** Sampling data of the studied *Dodonea viscosa* and percentage yield of the essential oil.

Voucher Number	Collection Places (Altitude)	Geographical Coordinates (GPS System)	Date	Oil Yield (%)
REU08438	Vincendo (67 m)	21°22′859 S 55°40′108 E	February 2009	0.10

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
