# Peer review of "Modified Clerodanes from the Essential Oil of Dodonea viscosa Leaves"

_molecules, 2020, doi:10.3390/molecules25040850_

Round 1

Reviewer 1 Report

The manuscript deals with a typical Natural Product (phytochemical) research. Leaves of a certain plant are extracted under a hydrodistillation process, and the extract is further analyzed utilizing a series of typical tools such as GC-MS, NMR, and other tools. Although the study is well-conceived and executed, there is no clear evidence about the innovation proposed as well as the importance worldwide of the investigated plant.

Author Response

Reviewer 1: Although the study is well-conceived and executed, there is no clear evidence about the innovation proposed as well as the importance worldwide of the investigated plant.

Author: As it is specified in the introduction of this paper, Dodonea viscosa has retained our attention not only because the plant possesses a great repute in folk medicine all around the world but also because it is registered un the French pharmacopoeia anf for this reason could be sold and consumed.

The innovation proposed is more in the method used (computational studies) to discuss about the stereochemistry of the isolated compounds.

Reviewer 2 Report

The paper entitled “Dodovisnoids H, I and J, three modified clerodanes from the essential
oil of Dodonea viscosa leaves” reported identification and characterization of essential oil composition obtained from Dodonea viscose leaves. Moreover three new clerodane-type diterpenes were characterized.

In my opinion, the paper cannot be accepted for publication. The main criticism concerns the compound dodovisnoid I (2). The authors report that it is a new compound but it was previously isolated from the same plant, Dodonaea viscosa, by Nui et al 2010 and namely dodovisate A.  (Niu, H. M., Zeng, D. Q., Long, C. L., Peng, Y. H., Wang, Y. H., Luo, J. F., ... & Zhao, F. W. (2010). Clerodane diterpenoids and prenylated flavonoids from Dodonaea viscosa. Journal of Asian natural products research12(1), 7-14.)

The comparison of the spectroscopic data seems to confirm the identity between 2 and the methyl dodovisate A, but there are difference in the optical rotation data. The authors report for 2 (and also for 3) 0 while for the methyl dodovosate A is  +56.4. Why?

Considering the structural correlation with dodovisate A also the names of compounds 1 and 3 should be changed,  only as example: dodovisate C for 1 and methyl iso-dodovosate A for 3. In fact the reported compounds do not seem related to dodovisnoids A-G (Zhang, L. B., Liao, H. B., Zhu, H. Y., Yu, M. H., Lei, C., & Hou, A. J. (2016). Antiviral clerodane diterpenoids from Dodonaea viscosaTetrahedron72(49), 8036-8041.)

Supplementary Materials are missing. The authors should report the spectra of the identified compounds.

Other comments

Because 1-3 are named bicyclo [5.4.0] undecane. The name refers to a tricyclic system for example with oxygen. As an example see Krishna, U. M., Srikanth, G. S. C., Trivedi, G. K., & Deodhar, K. D. (2003). Asymmetric oxidopyrylium-alkene [5+ 2] cycloaddition: Synthesis of enantiopure oxa-bridged bicyclo [5.4. 0] undecanes. Synlett, 2003 (15), 2383-2385.

In my opinion paragraph 2.3 on Biosynthetic pathway and the relative figure must be removed. Few  information can be reported in the text.

Table 1 can be removed and the relative information reported in the Material and Methods section

Author Response

Reviewer 2: In my opinion, the paper cannot be accepted for publication. The main criticism concerns the compound dodovisnoid I (2). The authors report that it is a new compound but it was previously isolated from the same plant, Dodonaea viscosa, by Nui et al 2010 and namely dodovisate A.  (Niu, H. M., Zeng, D. Q., Long, C. L., Peng, Y. H., Wang, Y. H., Luo, J. F., ... & Zhao, F. W. (2010). Clerodane diterpenoids and prenylated flavonoids from Dodonaea viscosa. Journal of Asian natural products research12(1), 7-14.).

Authors: We do apologize for this error. We have rectified and clarified this point in the revised manuscript.

Reviewer 2: The comparison of the spectroscopic data seems to confirm the identity between 2 and the methyl dodovisate A, but there are difference in the optical rotation data. The authors report for 2 (and also for 3) 0 while for the methyl dodovosate A is  +56.4. Why?

Authors: It was again a mistake: a “copy/paste” applied when writing the manuscript.

Reviewer 2: Considering the structural correlation with dodovisate A also the names of compounds 1 and 3 should be changed,  only as example: dodovisate C for 1 and methyl iso-dodovosate A for 3. In fact the reported compounds do not seem related to dodovisnoids A-G (Zhang, L. B., Liao, H. B., Zhu, H. Y., Yu, M. H., Lei, C., & Hou, A. J. (2016). Antiviral clerodane diterpenoids from Dodonaea viscosaTetrahedron72(49), 8036-8041.)

Authors: This has been taken into account in the new version.

Reviewer 2: Supplementary Materials are missing. The authors should report the spectra of the identified compounds.

Authors: A supplementary materials is now proposed.

Reviewer 2: Because 1-3 are named bicyclo [5.4.0] undecane. The name refers to a tricyclic system for example with oxygen. As an example see Krishna, U. M., Srikanth, G. S. C., Trivedi, G. K., & Deodhar, K. D. (2003). Asymmetric oxidopyrylium-alkene [5+ 2] cycloaddition: Synthesis of enantiopure oxa-bridged bicyclo [5.4. 0] undecanes. Synlett, 2003 (15), 2383-2385.

Authors: We do not understand very well this remark as for us a bicyclo[5.4.0] undecan refers to a bicyclic system. See for example “Clerodane diterpenoids from the roots of Portulaca pilosa.” Ohsaki A. et al. Phytochemistry, 30 (12), 4075-4077.

Reviewer 2: In my opinion paragraph 2.3 on Biosynthetic pathway and the relative figure must be removed. Few  information can be reported in the text.

Authors: This has been taken into account in the new version.

Reviewer 2: Table 1 can be removed and the relative information reported in the Material and Methods section

Authors:This has been taken into account in the new version.

Reviewer 3 Report

I reviewed the manuscript entitled: Dodovisnoids H, I and J, three modified 
clerodanes from the essential oil of Dodonea  viscosa leaves".

General comments:

This is an interesting research work related to the Dodovisnoids H, I and J, three modified clerodanes from the essential oil of Dodonea  viscosa leaves" The paper presents a comprehensive study described in a clear straightforward and detailed form. Also, the manuscript is well-written but it contains some omissions that must be clarified. To my opinion, is a good paper and it can be published after just minor corrections.

Abstract. Ok

Introduction.  Ok

Results and Discussion. Ok

 Material and methods:

 Line 334-355

The authors from where you obtained the method are not cited. It is important to clarify if the method selected was according to an author or if it was modify.

Line 360-362

The authors from where you obtained the method are not cited. It is important to clarify if the method selected was according to an author or if it was modify.

 Line 364 373

The authors from where you obtained the method are not cited. It is important to clarify if the method selected was according to an author or if it was modify.

Author Response

Reviewer 3: Line 334-355. The authors from where you obtained the method are not cited. It is important to clarify if the method selected was according to an author or if it was modify.

Authors: The general experimental procedure described was not selected according to an author. Through our experience, we have developed the methodology applied in this work.

Reviewer 3: Line 360-362. The authors from where you obtained the method are not cited. It is important to clarify if the method selected was according to an author or if it was modify.

Authors: The essential oil extraction was carried out according a standard method. Generally, it is not used to refer to an author when applying this method. See for example “Insecticidal activity of the leaf essential oil of Peperomia borbonensis MIQ. (Piperaceae) and its major cComponents against the melon fly Bactrocera cucurbitae (Diptera: Tephritidae)” Dorla E. et al. Chemistry and Biodiversity, 14(6), 2017.

Reviewer 3: Line 364 373. The authors from where you obtained the method are not cited. It is important to clarify if the method selected was according to an author or if it was modify.

Authors: The methodology applied in this work was developed in our laboratory and does not reproduced a methodology described in a published paper.

Round 2

Reviewer 1 Report

The revised manuscript introduced considerable improvements related to the original submission.  

Reviewer 2 Report

The authors replied to comments raised in the previous review and revised the manuscript according my suggestion. The manuscript could be accepted for publication in present form.